# ALGEBRAIC BIOINFORMATICS BASED ON MATRIX-OPERATOR ALPHABETS

## FOR ARTIFICIAL INTELLIGENCE

**Abstract**. In the field of artificial intelligence, scientists strive to replicate the features of living organisms, whose structures are inherited from generation to generation, i.e., they are linked to a system of bioinformation coding. It is known that protein sequences of amino acids are genetically inherited using coded messages in DNA and RNA molecules based on an alphabet of 4 nucleotides. However, as Nobel laureate T. Steitz emphasizes, all knowledge about these biomolecules, encoded in the genome in this biochemical alphabet, does not tell us, for example, how a butterfly flies or how a turtle, upon hatching from its egg, immediately begins to crawl towards the water, which requires the logically coordinated activity of millions of nerve and muscle cells. Thus, in modern science of biological inheritance, there is a lack of knowledge about a bioinformation system capable of ensuring the inheritance of cooperative phenomena of algorithmic behavior of body parts. These inherited logical forms of algorithmic behavior in biosystems necessitate the search for a corresponding algebraic-operator bioinformation system, based on suitable alphabets and coupled with nucleotide-alphabet bioinformatics. As a result of this search, the author has discovered two structurally related matrix-operator alphabets, each containing 4 operators, which form the basis for two effective types of algebraic-operator bioinformatics, termed cyclic and spiral types. They can be considered as components of a unified cyclo-spiral bioinformatics. The article is devoted to these alphabets and the types of bioinformatics based on them, offering new possibilities for modeling the inheritance of biological phenomena. A paradigm of code algebraic-operator Darwinism is formulated, according to which natural selection and the inheritance of the most survival-beneficial code combinations of alphabetic operators of algebraic bioinformatics play an important role in evolution. The features of operator algebraic bioinformatics are proposed to be considered in the development of genomorphic artificial intelligence.

**Keywords**: bioinformatics, quantum informatics, quantum logic, unitary Hadamard matrices, cycles, systolic processors, spirals, dual numbers, screw calculus, Fibonacci matrices, artificial intelligence.

## 1. Introduction

All genetically inherited physiological organs are structurally coupled with the bioinformation system of genetic texts in DNA and RNA molecules. Understanding the secrets of genetic informatics is crucial for the development of biotechnologies, artificial intelligence, medicine, etc. Information communication systems are built using a particular alphabet for forming information messages. For example, all computer programs rely on corresponding programming alphabets. The science of biological inheritance today is based on knowledge of the 4-nucleotide alphabets of DNA and RNA, i.e., on nucleotide-alphabet bioinformatics, which has contributed greatly to the understanding of proteins and nucleic acids. But, as Nobel laureate in chemistry, geneticist T. Steitz, emphasizes, all knowledge about the biochemical structure of proteins and nucleic acids encoded in the genome does not tell us, for example, how a butterfly flies [1]. And they do not tell us how turtles, hatching from an egg, immediately without any training begin to crawl towards the water using coordinated limb movements, which requires the logically coordinated activity of millions of their nerve and muscle cells. And they do not illuminate how a newborn infant emits a recognition cry and begins to suckle at the mother's breast, which also requires the logically coordinated activity of billions of nerve and muscle cells. From knowledge of nucleotide sequences in DNA/RNA, one cannot deduce the fact of inheritance of the geometric beauty of

biological forms (mollusk shells, etc.), repeated in bodies of very different biochemical composition and constructed through the spatiotemporal ordering of trillions of different molecules. Also, knowledge of the biochemical alphabet of 4 nucleotide types does not allow us to understand how one-dimensional nucleotide sequences in genomic DNA can encode the inherited three-dimensional forms of living bodies.

Consequently, in modern science of biological inheritance, there is a lack of knowledge about a bioinformation system capable of ensuring the inheritance of phenomena of coordinated algorithmic behavior of body parts. These inherited logical forms of collective behavior in biosystems require, for their modeling, the search for a corresponding operator bioinformation system based on a suitable alphabet. This suggests that, alongside nucleotide-alphabet bioinformatics, a mutually complementary operator-alphabet bioinformatics operates in living systems. This hidden algebraic-operator bioinformatics and its alphabet are presumably coupled with quantum mechanics and quantum informatics, since genetic molecules belong to the microcosm of quantum mechanics. In this search, special attention should be paid to unitary operators, which serve as the logical gates for all computations in quantum computing [2] and, in quantum mechanics, describe the evolution of closed quantum systems. It is worth noting that the search for effective approaches to modeling logically organized biological processes is ongoing worldwide, including appeals to quantum mechanics and quantum informatics (see, for example, the collection [3]). This article is primarily devoted to the author's discovery of a genetic alphabet consisting of 4 unitary Hadamard operators and the development of operator quantum-logical bioinformatics based on it for mathematical modeling of the inheritance features of the structure and behavior of multicomponent body parts, including the features of inheriting a multitude of temporally coordinated cyclic processes.

The founder of quantum informatics, Yu.I. Manin, introduced the concept of a quantum computer in his book precisely while analyzing the features of high-speed information processing by "*genetic automata*" in chromosomal DNA, prophetically pointing out the important role of unitary operators and tensor products [4, p.15]: "*A quantum automaton must be abstract: its mathematical model should use only the most general quantum principles, without predetermining physical realizations. Then the model of evolution is a unitary rotation in a finite-dimensional Hilbert space, and the model of virtual separation into subsystems corresponds to the decomposition of the space into a tensor product. Somewhere in this picture, interaction, traditionally described by Hermitian operators and probabilities, must find its place.*" Thus, the very birth of quantum informatics, so promising for the development of artificial intelligence and quantum-logical biology, occurred due to the desire to understand the features of genetic informatics. The data in this article are consistent with the quoted prophecy of Yu.I. Manin.

This article also presents the discovery of an algebraic-biological connection between the aforementioned operator bioinformatics, based on the alphabet of 4 unitary Hadamard (2x2)-matrices, and a structurally related operator bioinformatics, also based on an alphabet of 4 (2x2)-matrix operators that are not unitary. The modeling approach based on this second operator alphabet reveals a coupling of inheritance biology with screw calculus, Fibonacci matrices, and orderly growing spiral structures, which are widespread in living nature and correspond, in particular, to phyllotaxis laws of morphogenesis. The aim of the article is to present the results of

studying the features and applications of these two named operator alphabets of genetics and the bioinformation systems based on them. It is noted that initially the author discovered the operator alphabet of 4 unitary Hadamard matrices, which later led to the identification of a related second (non-unitary) operator alphabet.

## 2. Genetic Matrices of DNA Nucleotide Alphabets and the Alphabet of 4 Unitary Hadamard Operators for Modeling Heredity Phenomena

Amino acid sequences of proteins are inherited thanks to the DNA alphabet of 4 nucleotides: G, A, T, C. These molecules have specific binary molecular features that are crucial for their biological functions:

1) Two of these nucleotides are purines (A and G), having two rings in their molecule, while the other two (C and T) are pyrimidines, containing one ring. This yields a binary representation (binary sub-alphabet):     C = T = 0, A = G = 1;

2) Two of these nucleotides are keto-molecules (T and G), and the other two (C and A) are amino-molecules, yielding a binary representation: C = A = 0, T = G = 1;

3) The pairs of complementary nucleotides A-T and C-G are linked by 2 and 3 hydrogen bonds (referred to in genetics as weak and strong hydrogen bonds), respectively, yielding a binary representation: C = G = 0,  A = T = 1.

These binary features of the DNA (and RNA) nucleotide alphabet in all living organisms are summarized in Table 1, which shows the distribution of these traits.

**Table 1**. Distribution of binary molecular traits in the DNA alphabet of 4 nucleotides G, A, T, C, with binary oppositions denoted by symbols +1 and -1. The fourth row of the table uses the symbol +1 to indicate the presence of benzene rings in the molecules of all four nucleotides.

| Molecular traits and their symbols | G | A | T | C |
|---|---|---|---|---|
| Pyrimidines +1, purines -1 | -1 | -1 | +1 | +1 |
| Amino-nucleotides +1, keto-nucleotides -1 | -1 | +1 | -1 | +1 |
| Complementarity with 3 or 2 hydrogen bonds: +1, -1 | +1 | -1 | -1 | +1 |
| The presence of benzene rings +1 | +1 | +1 | +1 | +1 |

      On the right side of this phenomenological Table 1, a symmetric Hadamard matrix of the fourth order emerges, which is a real Hermitian matrix, and its quadrants are occupied by 4 types of second-order Hadamard matrices. Recall that Hadamard matrices $H_n$ of order n are square matrices composed of elements +1 and -1 satisfying the criterion $H_n \times H_n^T = nE$, where E is the identity matrix of order n. Fig. 1 shows these 4 second-order Hadamard matrices from Table 1, normalized by the factor $2^{-0.5}$, traditionally used in quantum mechanics and quantum computing to render the matrices unitary. Hadamard matrices possess many remarkable properties and applications.

$$H_C = 2^{-0.5} \begin{vmatrix} 1, -1 \\ 1, \; 1 \end{vmatrix} ; \quad H_T = 2^{-0.5} \begin{vmatrix} 1, \; 1 \\ 1, -1 \end{vmatrix} ; \quad H_G = 2^{-0.5} \begin{vmatrix} 1, \; 1 \\ -1, 1 \end{vmatrix} ; \quad H_A = 2^{-0.5} \begin{vmatrix} -1, 1 \\ 1, \; 1 \end{vmatrix}$$

**Fig. 1** – Four unitary Hadamard matrices $H_C$, $H_A$, $H_T$, and $H_G$.

The identification of this connection between the molecular nucleotide alphabet of DNA and these 4 unitary Hadamard matrices is important due to the significance of unitary transformations (unitary operators) for quantum mechanics, quantum computing, signal processing engineering, etc. Unitary transformations preserve vector lengths and inner products (preserve the metric), representing operators of rotation and mirror reflection. Unitary matrices satisfy the criterion: the product of a unitary matrix and its transposed conjugate equals identity. Unitary transformations with real components are also called orthogonal transformations, but we will use their more general name, "unitary transformations," in this article, as they are better known across various scientific fields. In quantum mechanics, unitary transformations describe the evolution over time of closed quantum systems, and in quantum mechanics (unlike classical mechanics), observable quantities are represented not by numbers but by operators. In quantum computers, all computations are performed precisely on the basis of unitary operators, acting as gates, and any unitary operator can be used as a gate in quantum computing [2]. Hadamard matrices are fundamental building blocks of quantum computers, providing qubit superposition and being a key element of quantum parallelism in the quantum Fourier transform and other quantum algorithms.

We will call the four unitary operators $H_C$, $H_A$, $H_T$, and $H_G$ (Fig. 1) genetic Hadamard gates and consider them as the sought-after quantum-operator alphabet, forming the basis of a quantum-logical bioinformation system for modeling inherited phenomena of cooperative biomechanics. These 4 matrices cyclically transform into each other upon repeated 90-degree rotations. This same set of 4 Hadamard matrices is also revealed when analyzing molecular-genetic informatics from other approaches. For example, the binary molecular traits indicated in Table 1 allow placing these 4 alphabetic nucleotides into the four possible positions of (2×2)-matrices, whose rows are numbered by binary numbers 0 and 1 from one binary sub-alphabet (e.g., the sub-alphabet of amino-nucleotides and keto-nucleotides: C = A = 0, T = G = 1), and columns by binary numbers 0 and 1 from another (e.g., the sub-alphabet of pyrimidines and purines: C = T = 0, A = G = 1) (Fig. 2). In such a matrix, all 4 nucleotides C, A, T, G occupy individual positions, since from the perspective of these two sub-alphabets, nucleotide C gets binary index **00**, nucleotide A – binary index **01**, nucleotide T – binary index **10**, nucleotide G – binary index **11** (the first binary symbol of the nucleotide comes from its row index, and the second from its column index).

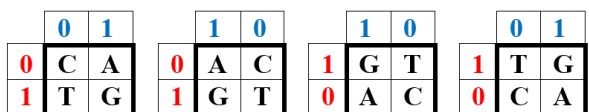

**Fig. 2** – All 4 possible arrangements of the 4 nucleotides C, A, G, T in genetic (2×2)-matrices, whose rows are indexed by binary numbers of the sub-alphabet of amino-nucleotides (C, A) and keto-nucleotides (T, G), and columns by binary numbers of the sub-alphabet of pyrimidines (C, T) and purines (A, G).

However, the DNA alphabet of 4 nucleotides contains yet another binary opposition inherent in nature: during the transition from DNA to RNA, only one nucleotide, T (thymine), is replaced by U (uracil), while other nucleotides C, A, G remain unchanged. This binary opposition is expressed by the representation: C = A = G = +1, T = -1. Taking this binary representation into account transforms the 4 symbolic genetic matrices from Fig. 2 into the 4 numerical matrices in Fig. 3, which are the same Hadamard matrices discussed above in connection with Fig. 1 and

which become the unitary matrices $\mathbf{H_C}$, $\mathbf{H_A}$, $\mathbf{H_T}$, and $\mathbf{H_G}$ upon the same normalization by multiplication by the factor $2^{-0.5}$.

**Fig. 3** – The set of 4 numerical Hadamard matrices arising from the 4 symbolic genetic matrices in Fig. 2 by considering the genetic binary opposition of nucleotide T to nucleotides A, C, G, giving the binary representation: C = A = G = +1, T = -1.

Briefly, it should be mentioned that analogous Hadamard (2×2)-matrices are also revealed in other types of analysis of structural and stochastic features of genetic informatics. For instance, considering Chargaff's second rule [5] concerning the ratios of probabilities of the 4 nucleotides in all long single-stranded DNAs (length > 100 kb) of higher and lower organisms leads to the real Hermitian probability matrix $\mathbf{W_D} = [0.5, 0.5; 0.5, 0.5]$. This genetic Hermitian matrix $\mathbf{W_D}$ is doubly stochastic: the sum of elements in each row and each column equals one. Regarding doubly stochastic matrices, the following theorem is known [6]:

- If a matrix $\mathbf{V} = \|v_{ij}\|^n$ is unitary, then the matrix $\mathbf{W} = \|w_{ij}\|^n$, where $w_{ij} = |v_{ij}|^2$, is doubly stochastic.

According to this theorem, the aforementioned doubly stochastic genetic probability matrix $\mathbf{W_D}$ corresponds to the 4 unitary alphabetic Hadamard matrices $\mathbf{H_C}$, $\mathbf{H_A}$, $\mathbf{H_T}$, and $\mathbf{H_G}$ from Fig. 1: squaring the absolute values of all components of each of these unitary matrices generates the doubly stochastic matrix $\mathbf{W_D}$. It is noted that these same 4 unitary matrices $\mathbf{H_C}$, $\mathbf{H_A}$, $\mathbf{H_T}$, and $\mathbf{H_G}$, but taken with a minus sign, are considered trivial analogues and are not treated separately. The totality of results from analyzing molecular-genetic informatics suggests a fundamental bioinformation significance of these alphabetic unitary Hadamard matrices for the bioinformation coding of inherited biological phenomena, especially those related to the ideology of closed quantum-like biosystems. This includes inherited cyclic and biorhythmic phenomena, which, as will be shown below, are effectively modeled using these unitary matrices.

One of these 4 genetic unitary Hadamard operators (gates) – $\mathbf{H_T}$ – has long been used in quantum computers for fundamental qubit operations, being a key element in many quantum algorithms, including the Deutsch-Jozsa algorithm and Shor's algorithm. This Hadamard gate provides the superposition principle in quantum algorithms for working with quantum entanglement, thereby demonstrating quantum supremacy – their significantly more efficient performance compared to known classical algorithms [2].

Among the 4 genetic Hadamard gates, two gates, $\mathbf{H_A}$ and $\mathbf{H_T}$, are mirror reflection operators. Raising them to integer powers generates corresponding cyclic groups of unitary operators with a period of 2. The other two unitary matrices ($\mathbf{H_C}$, $\mathbf{H_G}$) are rotation operators ($\mathbf{H_C}$ counterclockwise, $\mathbf{H_G}$ clockwise) and matrix representations of the complex number $Z = (1+i) \cdot 2^{-0.5}$, where i is the imaginary unit ($i^2 = -1$). Repeated exponentiation of these unitary matrices $\mathbf{H_C}$ and $\mathbf{H_G}$ to integer

powers (positive and negative) generates cyclic (with period 8) groups of unitary operators, which are matrix representations of complex numbers (1).

$$H_C^n = H_C^{n+8}, \quad H_G^n = H_G^{n+8}, \quad H_A^n = H_A^{n+2}, \quad H_T^n = H_T^{n+2} \tag{1}$$

Moreover, any of the unitary matrices $H_C$ or $H_G$ can be represented as the product of k unitary matrices that are the k-th roots of it. In other words, the action of a single unitary operator, e.g., $H_C$, can be represented as the action of a sequence of k finer unitary operators $H_C^{1/k}$. Thus, in a matrix-vector approach, any large transformation in a system resulting from the action of such a whole operator $H_C$ can be represented as consisting of a sequence of arbitrarily small transformations from the action of the corresponding sequence of unitary operators $H_C^{1/k}$, allowing the modeling of quasi-continuous transformations in cyclic processes. Additionally, it is noted that raising the genetic gate $H_C$ or $H_G$ to a power representing a cyclic function of time stepwise enables the modeling of quasi-continuous cyclic bioprocesses as vector sequences of their stepwise states.

In quantum logic, projectors (projection operators) play an important role. In light of this, it is noted that the unitary Hadamard matrices in operators $H_C$ or $H_G$ are sums of sparse matrices which are shown in expression (2) in parentheses and which represent projectors $P_s$, satisfying the projector criterion $P_s^2 = P_s$:

$$H_C = 2^{-0.5} \bullet [1\ -1;\ 1\ 1] = 2^{-0.5} \bullet ([1, 0;\ 1, 0] + [0, -1;\ 0, 1]),$$

$$H_G = 2^{-0.5} \bullet [1\ 1;\ -1\ 1] = 2^{-0.5} \bullet ([1, 0;\ -1, 0] + [0, 1;\ 0, 1]) \tag{2}$$

Many genetically inherited biological structures in organisms are obviously related to unitary transformations of rotations and mirror reflections. For example, the kinematic scheme of the human body and its locomotion is based on unitary transformations of rotations in joints (there are about 300 joints in the human body) and the mirror symmetry of the left and right halves of the body. Human motor activity boils down to the nervous system's skillful control of ensembles of these unitary transformations in body kinematics, which is coupled with the genetically inherited ability of the nervous system to operate with unitary transformations. Moreover, the human representation of his/her body schema is innate: people born without limbs, having no personal experience of using them, nonetheless feel them as really existing, with phantom pains in them [7, 8].

When studying human sensorimotor features, it must be considered that the genetically inherited nervous system in its structural organization is related to genetic structures. A person sees the world through probabilities in the statistical signal streams from retinal neurons (containing millions of receptor cells) and other sensory organs. Norbert Wiener, the father of cybernetics, asserted: "*genetic memory – the memory of our genes – is essentially determined by complexes of nucleic acids... there are reasons to think that the memory of the nervous system has the same nature*" [9, 10].

Another example of the biological importance of unitary transformations is the construction of the complex three-dimensional shape of proteins, i.e., protein folding. These shapes are built on

unitary transformations of rotations of segments of protein molecules relative to each other around relatively strong carbon-carbon bonds. Many biotechnologies are associated with knowledge about proteins.

The author proposes to consider and use the family of 4 genetic unitary Hadamard operators $H_C$, $H_A$, $H_T$, and $H_G$ (Fig. 1) as a basic genetic quantum-logical alphabet for developing a theory of a quantum-logical bioinformation system. This system would allow modeling the inheritance of algorithmically organized biological structures and phenomena, primarily of a cyclic and rhythmic nature, as described in the preprint [11]. Consequently, this type of algebraic-operator bioinformation system is briefly termed "cyclic bioinformatics." Within its framework, unitary matrix representations of Hamilton's quaternions and biquaternions, projection operators, etc., arise and are studied [11].

Here, the distinctive features of quantum logic [12, 13] should be explained. From a formal point of view, quantum logic is an algebraic system for describing, using quantum gates, how qubits work and interact and how to extract information from them. In quantum logic, "logic" is not about reasoning but about the mathematical description of states and operations. Quantum logic can be formulated as a modified version of propositional logic. For comparison, recall that classical Boolean logic is a set of logical rules (AND, OR, NOT, etc.) describing how bits (0 or 1) can be combined and transformed according to the laws of Boolean algebra, with its key principle of distributivity and "true" or "false" statements. Quantum logic deals not with "true/false" statements but with questions posed to a quantum system. The answer to such a question is a probability value obtained upon measurement. Logical operations are replaced by quantum gates (unitary operations): NOT becomes the X gate, and entirely new operations with no classical analogues appear, such as the Hadamard gate, which creates superposition. Quantum logic works with qubits, vectors, and matrices, not with sets. Its mathematical foundation is the theory of Hilbert spaces and projective and unitary operators. The state space of a quantum system is described by vectors, and the rotations of these vectors serve as logical operations. Distributivity is absent in quantum logic, which is considered its key distinction from Boolean logic. Quantum logic is a branch of logic necessary for reasoning about propositions that account for the principles of quantum theory. It was founded by the work of G. Birkhoff and J. von Neumann [14], who attempted to reconcile the inconsistency of classical logic with the facts about measurements in quantum mechanics and saw in quantum logic a possible foundation for physics.

The unitary Hadamard matrices of the quantum-operator alphabet of bioinformatics $H_C$, $H_A$, $H_T$, and $H_G$ (Fig. 1) and many types of their combinations into higher-order unitary matrices form – upon repeated exponentiation – cyclic groups of operators with different periods, used for modeling cyclic sequences of states of quantum-like systems. The algebraic-geometric apparatus arising here is intended primarily for quantum-logical modeling of the multitude of genetically inherited cyclic and hypercyclic biostructures in genetic biomechanics. It is emphasized that the author's quantum-logical approach to inherited cyclic and hypercyclic biostructures, based on the alphabet of genetic Hadamard gates and the mathematical apparatus of quantum-logical biology, fundamentally differs from the well-known biochemical concept of catalytic cycles and hypercycles [15].

Worldwide research in genetic informatics largely relies on the fundamental fact of the existence in the DNA of all organisms of the molecular alphabet of 4 nucleotides C, G, A, T and its extensions into alphabets of 16 duplets, 64 triplets, etc. In parallel with this molecular alphabet of 4 nucleotides, it is now possible and necessary to work in bioinformatics and genetic biomechanics with the operator alphabet of 4 genetic Hadamard gates, i.e., an alphabet of a fundamentally new type: a quantum-logical operator-code alphabet of 4 unitary matrices $H_C$, $H_A$, $H_T$, and $H_G$. On its basis, interconnected sets of higher-order unitary operators arise, as well as their cyclic power groups, providing new approaches for modeling, primarily, inherited cyclic and biorhythmic phenomena, as expounded by the author in the preprint [11]. These genetic unitary Hadamard matrices are associated with corresponding complete orthogonal systems of Walsh functions. The latter form the basis of special spectral signal analysis in digital informatics and are associated with cyclic Gray codes, logical holography, Walsh antennas, and also the fractal Hilbert curve, which is known to correspond to the spatial packing of chromatin in the human genome [16]. The connection of Hadamard matrices with the listed areas is described in our works [17-19]. The mathematical properties of the alphabet of 4 genetic Hadamard gates and the sets of unitary operators built upon them are subject to systematic study.

## 3. A Related Alphabet of 4 Non-Unitary Operators for the Bioinformatics of Spiral Structure Inheritance

The previous section presented elements of the theory and applications of a bioinformation system based on the genetic alphabet of 4 unitary Hadamard operators. In quantum mechanics, unitary operators describe the evolution of closed quantum systems. Accordingly, this system can be conditionally termed a "unitary bioinformation system," useful primarily for modeling the biological inheritance of a wide class of algorithmic cyclic and biorhythmic ensembles [11].

However, biology also knows the inheritance of structures that, in principle, cannot be described by this unitary-operator bioinformatics of closed quantum systems and should therefore be attributed to the category of bioinformatics of open (non-closed) quantum-like systems. This refers, for example, to the abundance of spiral and screw structures at all levels of organization of living bodies. Spirals are represented in vessels, bones, nerves, the cochlea of the ear, tendons and ligaments, alpha-helices of proteins, collagen, flagella and cilia of the motor apparatus of bacteria and many unicellular organisms, mollusk shells, animal horns, the cellular organization of the embryo (zygote), etc. In particular, the heart is formed by a single muscle twisted into a spiral, and the propulsion of blood through vessels is ensured by changing the degree of muscular twisting (similar to a laundress twisting out wet laundry to squeeze out the water); the principle of spiral twisting is repeated at all levels of the macro- and microstructure of the heart. Due to the almost total spiral nature of inherited biological configurations, all fluid flows in the body – blood, lymph, urine – have a spiral character. It is no wonder that Goethe called spirals "symbols of life," and a book with the telling title "The Curves of Life" is dedicated to them [20]. Biological spirals can serve as bio-antennas for emitting and receiving circularly polarized electromagnetic waves, forming highly efficient bio-antenna arrays that determine communication and other functional features in many organisms; this is systematically considered with many biological examples in publications on the doctrine of energy-information evolution based on bio-antenna arrays [21, 22].

Data on biological spirals are used in the bionics of spiral structures [23]. Spiral and fractal-spiral antennas are widely used in communication technology due to their generation of circularly polarized electromagnetic waves, which have many useful applications and are associated with the theme of inherited chirality (enantiomorphism) of biological structures and bio-antenna arrays. Characteristically, inherited biological spirals often obey the known laws of phyllotaxis, which connect them with Fibonacci numbers from the recurrent sequence $F_{n+2} = F_n + F_{n+1}$, starting with $F_0 = 0$, $F_1 = 1$ (Table 2). The study of spiral phyllotaxis and the features of its modular (block) structure has been conducted worldwide for about a century and a half, constituting one of the most famous sections of mathematical biology, to which many hundreds of publications are devoted. In this section, considerable attention is also given to the golden ratio or "divine proportion" $\varphi = (1+\sqrt{5})/2 = 1.618\ldots$, which is related to the Fibonacci number series and has served as a mathematical symbol of self-reproduction since the Renaissance. Many authors publish data on the manifestation of the golden ratio in various inherited physiological systems: cardiovascular, respiratory, locomotor, electrical activity of the brain, psychophysiological, etc. (see review data on the golden ratio, for example, in [17, section 2.2]).

**Table 2**. Initial segment of the Fibonacci sequence $F_{n+2} = F_n + F_{n+1}$.

| n | 0 | 1 | 2 | 3 | 4 | 5 | 6 | 7 | 8 | 9 | 10 | 11 | 12 | ... |
|---|---|---|---|---|---|---|---|---|---|---|----|----|----|-----|
| $F_n$ | 0 | 1 | 1 | 2 | 3 | 5 | 8 | 13 | 21 | 34 | 55 | 89 | 144 | ... |

Many spiral configurations of growing living bodies, for example, phyllotaxis spirals, consist of algorithmically repeating bodily modules, reminiscent of the ancient principle "like begets like," represented in the organization of the DNA double helix, etc. [24]. Such algorithmic cyclomeric construction of biological bodies in plants, mollusk shells, and many other types of organisms is accompanied by the realization of A.V. Shubnikov's biosymmetries of similarity and their generalizations in the form of conformal-geometric (locally-like or Möbius) biosymmetries [22, 25]. The fact of inheritance of families of ordered biological spirals at all levels of living organization suggests the existence in living nature of another operator bioinformation system based on an algebraic-operator alphabet suitable for encoding and inheriting spiral and screw structures, including phyllotaxis. The operators of such an alphabetic bioinformatics should allow modeling the ensemble or modular multiplication of spiral configurations, including those associated with Fibonacci numbers.

We will now proceed to describe such a "spiral" bioinformation system based on a genetic alphabet of 4 matrix operators, closely related to the described genetic alphabet of 4 unitary Hadamard matrices: both alphabets consist of related sets of 4 matrices that are trivial algorithmic modifications of each other. This alphabetic algebraic kinship allows both bioinformation systems to function in a mutually coordinated manner.

Let's return to Table 1, where the distribution of binary-opposition molecular traits in the DNA alphabet of 4 nucleotides is presented based on the binary numbers +1 and -1, revealing a connection with the alphabet of Hadamard matrices. But the same distribution of binary-opposition traits in the nucleotide DNA alphabet can be represented based on binary elements 1

and 0, replacing element -1 with element 0 in Table 1. The result of this algorithmic substitution is presented in Table 3.

**Table 3**. Binary distribution of molecular traits in the DNA alphabet of 4 nucleotides G, A, T, C, with binary oppositions denoted by symbols +1 and 0. The fourth row of the table uses the symbol +1 to indicate the presence of benzene rings in the molecules of all four nucleotides.

| Molecular traits and their symbols | G | A | T | C |
|---|---|---|---|---|
| Pyrimidines +1, purines -1 | -1 | -1 | +1 | +1 |
| Amino-nucleotides +1, keto-nucleotides -1 | -1 | +1 | -1 | +1 |
| Complementarity with 3 or 2 hydrogen bonds: +1, -1 | +1 | -1 | -1 | +1 |
| The presence of benzene rings +1 | +1 | +1 | +1 | +1 |

On the right side of this phenomenological Table 3, a matrix is shown, all four quadrants of which have significant algebraic value. Let us denote the 4 (2x2)-matrices in these quadrants as $D_C$, $D_A$, $D_T$, $D_G$ (in some analogy with the previously considered alphabetic unitary matrices $H_C$, $H_A$, $H_T$, $H_G$). The lower right quadrant $D_G = [0, 1; 1, 1]$ is the well-known Fibonacci matrix $F$, whose exponentiation to integer powers n generates a family of matrices whose elements are all Fibonacci numbers $F_n$ (Table 2), represented in the biological laws of spiral phyllotaxis (Fig. 4).

$$F = \begin{vmatrix} 0, & 1 \\ 1, & 1 \end{vmatrix} ; \qquad F^2 = \begin{vmatrix} 1, & 1 \\ 1, & 2 \end{vmatrix} ; \qquad F^n = \begin{vmatrix} F_{n-1}, & F_n \\ F_n, & F_{n+1} \end{vmatrix}$$

**Fig. 4** - The Fibonacci matrix $F = D_G$ and its powers.

In the two quadrants along the second diagonal of the (4x4)-matrix in Table 3 are the (2x2)-matrices $D_A = [1, 1; 0, 1]$ and $D_T = [1, 0; 1, 1]$. These are matrix representations of hypercomplex dual numbers, which form the basis of screw calculus, a generalization of vector calculus associated with the names of W. Clifford, E. Study, A.P. Kotelnikov, and others [26]. It provides a compact description of the screw motion (rotation + translation along the rotation axis) of rigid bodies in our physical three-dimensional space; screw calculus is coupled with the theory of electromagnetic waves and photon helicity, and has numerous applications in robotics, manipulator control, imitation learning, computer vision, computer graphics, information storage and processing (including quantum informatics), antenna technology, holography recording screw wavefronts (in connection with optical vortices), automatic differentiation, etc. Dual numbers are also used in machine learning in connection with automatic differentiation, a key technology in modern neural networks [27]. A dual number is a hypercomplex number of the form $a + \varepsilon b$, where $a$ and $b$ are real numbers, and $\varepsilon$ is the dual imaginary unit or abstract element whose square is zero ($\varepsilon^2 = 0$), but which itself is not zero [28, 29]. Any dual number is uniquely determined by such a pair of numbers $a$ and $b$, where $a$ is called the real part and $b$ the dual part. The set of all dual numbers forms a two-dimensional commutative associative algebra with identity over the field of real numbers. Fig. 5 shows the matrix representation of a dual number as the aforementioned (2x2)-matrices $D_A$ and $D_T$, each being the sum of two sparse matrices, the first of which is the identity matrix representing the real unit, and the second represents the imaginary unit of the dual number $\varepsilon$ (thus, there are two different matrix representations of this imaginary unit $\varepsilon$ here). Each of these sets of two matrices is closed under multiplication and corresponds to the multiplication table of the algebra of dual numbers, shown in Fig. 5.

$$D_A = \begin{vmatrix} 1,1 \\ 0,1 \end{vmatrix} = \begin{vmatrix} 1,0 \\ 0,1 \end{vmatrix} + \begin{vmatrix} 0,1 \\ 0,0 \end{vmatrix} = E + \varepsilon \longrightarrow$$

| * | E | ε |
|---|---|---|
| E | E | ε |
| ε | ε | ε |

$$D_T = \begin{vmatrix} 1,0 \\ 1,1 \end{vmatrix} = \begin{vmatrix} 1,0 \\ 0,1 \end{vmatrix} + \begin{vmatrix} 0,0 \\ 1,0 \end{vmatrix} = E + \varepsilon \longrightarrow$$

| * | E | ε |
|---|---|---|
| E | E | ε |
| ε | ε | ε |

**Fig. 5** - Decomposition of the (2x2)-matrices $D_A$ and $D_T$ (from the quadrants along the second diagonal of the genetic (4x4)-matrix in Table 3) shows they are matrix representations of dual numbers; the multiplication table of their algebra is shown on the right.

What is additionally interesting that the product of these two genetic matrix representations of the dual numbers $D_A$ and $D_T$ (Fig. 6) yields the Fibonacci matrix $F$ to the second power (shown in Fig. 4), that is, these genetic operators of dual numbers are structurally coupled with the Fibonacci patterns of inherited phyllotaxis and with the matrix $D_G = F$.

$$D_T * D_A = \begin{vmatrix} 1,0 \\ 1,1 \end{vmatrix} * \begin{vmatrix} 1,1 \\ 0,1 \end{vmatrix} = \begin{vmatrix} 1,1 \\ 1,2 \end{vmatrix} = F^2$$

**Fig. 6** - The product of the alphabetic-genetic matrices of dual numbers $D_T$ and $D_A$ yields the square of the Fibonacci matrix $F^2$ from Fig. 4.

The matrix $D_C = [0, 0; 0, 1]$, located in the upper left quadrant of the (4x4)-matrix in Table 2, is a projection operator, since it satisfies the criterion for such operators: its square equals itself. As is known, projection operators in screw calculus are a powerful analytical and geometric tool for decomposition. They allow: to dissect complex spatial motion or force action into simple, intuitively understandable components (rotation, translation along and across the axis); to work with invariant subspaces, simplifying mathematical derivations; to visually interpret solutions of problems in mechanics and theory of mechanisms. Without projection operators, screw calculus would remain merely a compact form of notation; with them, it becomes an effective apparatus for deep geometric analysis. Incidentally, it is noted that the product of this genetic projector with the Fibonacci matrix raised to various integer powers generates a matrix operator whose non-zero elements are pairs of adjacent Fibonacci numbers (Fig. 7). The manifestation of precisely pairs of adjacent numbers from the Fibonacci sequence is characteristic of phyllotaxis phenomena, for example, for the numbers of left and right spiral seed arrangements in a sunflower head.

$$D_C * D_G^n = \begin{vmatrix} 0,0 \\ 0,1 \end{vmatrix} * \begin{vmatrix} 0,1 \\ 1,1 \end{vmatrix}^n = \begin{vmatrix} 0,\ 0 \\ F_n,\ F_{n+1} \end{vmatrix}$$

Fig. 7 - The product of the alphabetic projector $D_C$ and the alphabetic Fibonacci matrix $D_G$ raised to the power n generates a matrix operator whose non-zero components are adjacent members of the Fibonacci series (Table 1), similar to the pairs of adjacent numbers in inherited phyllotaxis phenomena.

This set of 4 matrix operators $\mathbf{D_C}$, $\mathbf{D_A}$, $\mathbf{D_T}$, $\mathbf{D_G}$ is considered by the author as the operator alphabet of a bioinformation system, conditionally called the spiral-operator bioinformation system (or briefly, "spiral bioinformatics") for modeling the biological inheritance of spiral structures using screw calculus. It interacts with the aforementioned unitary-operator bioinformation system ("cyclic bioinformatics"), based on the related alphabet of 4 unitary Hadamard operators $\mathbf{H_C}$, $\mathbf{H_A}$, $\mathbf{H_T}$, $\mathbf{H_G}$. The combined mathematical apparatus of cyclic and spiral bioinformation systems allows, in particular, modeling electromagnetic waves with their polarization and other features, which play an important role in biological organization, including communication between elements of a living body.

Additionally, it is found that if we return to Figs. 2 and 3, showing four variants of the possible arrangement of 4 nucleotides in genetic (2×2)-matrices, and algorithmically replace the element -1 with element 0 in the 4 binary matrices in Fig. 3, we obtain numerical matrices (Fig. 8), again associated with Fibonacci matrix and dual numbers of screw calculus, and hence with spiral-operator bioinformatics.

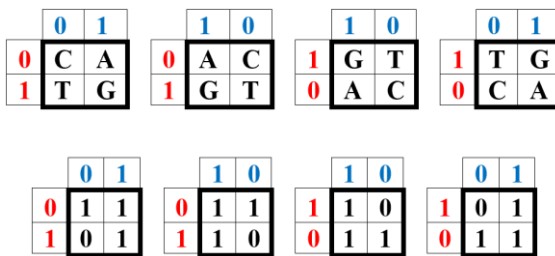

**Fig. 8** - Result of transforming the 4 genetic Hadamard matrices from Fig. 3 by algorithmically replacing the element -1 with 0.

Indeed, the first and third numerical matrices in Fig. 8 coincide with the matrices $\mathbf{D_A}$ and $\mathbf{D_T}$ discussed above, representing dual numbers of screw calculus. And the second and fourth numerical matrices in Fig. 8 are two variants of the Fibonacci matrix $\mathbf{F} = \mathbf{D_G}$ (Fig. 4), associated with inherited phyllotaxis spirals.

The action of these alphabetic operators $\mathbf{D_C}$, $\mathbf{D_A}$, $\mathbf{D_T}$, $\mathbf{D_G}$ of spiral bioinformatics, as well as their combinations into block matrices and tensor families of operators on vectors, certainly deserves systematic study in connection with the importance of the emerging possibilities for modeling inherited biological phenomena. In particular, identifying the structural connection of the molecular-genetic system with screw calculus allows using those aspects of screw calculus that are coupled with the operators of spiral bioinformatics for the purpose of modeling and comparative analysis of the multitude of bioinformatically inherited spiral forms, for example, the screw-like three-dimensional configurations of mollusk shells, long used in the theme "beauty of forms in nature" [30]. As is known, screw calculus allows effective modeling of spiral biological forms, and the results of such modeling can be visualized using publicly available computer tools like Desmos Graphing Calculator (3D), GeoGebra 3D Graphing Calculator, Wolfram|Alpha, etc.

## 4. On Unified Cyclo-Spiral Bioinformatics

Many inherited biological phenomena are structurally organized in such a way that they bear the features of logically coordinated cyclic and spiral processes. Examples include the growth cyclic processes of spiral phyllotaxis in plants and other organisms; the pulsating spiral movement of blood in the cardiovascular system; the cyclic limb movements in the locomotion of animal organisms (including humans and centipedes), most effectively modeled by screw calculus operators, etc. A striking example of the logical interaction of cyclic and spiral bioinformatics is provided by the unicellular organism *Mixotricha paradoxa* (https://en.wikipedia.org/wiki/Mixotricha_paradoxa): it moves due to 250 thousand spiral bacteria *Treponema spirochetes* located on its surface, whose spiral flagella work in concert as a single unit, ensuring the purposeful movement of the organism in desired directions and speeds through logical interactions and logical control (Fig. 9).

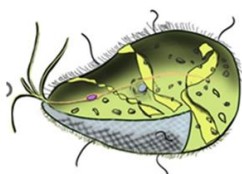

**Fig. 9** - The unicellular organism *Mixotricha paradoxa* moves purposefully due to the coordinated beating of the spiral flagella of 250 thousand bacteria on its surface (image from site https://commons.wikimedia.org/wiki/File:Mixotricha_paradoxa_color_and.png, permitting its copying and distribution under the "Creative Commons Attribution-Share Alike 4.0 International license").

Moreover, this organism has only 4 of its own spiral flagella, the movements of which can set (encode) the coordinated movements of hundreds of thousands of bacterial spiral flagella on its surface for the aforementioned purposeful movement of the entire organism. In other words, these 4 own flagella can be considered as members of a specialized 4-element alphabet, whose movement parameters serve as information signals for the algorithmic specification of the logically coordinated beats of those thousands of bacterial flagella. The physical mechanism of this bioinformation exchange is based on electromagnetic waves and the aforementioned bio-antenna arrays for emitting and receiving these waves using resonance principles. A spiral antenna differs from other antennas with directional radiation primarily in that its radiation field has circular polarization. Moreover, as is known, screw calculus provides an ideal, invariant, and geometrically meaningful language for the theory of electromagnetic waves, revealing the fundamental unity of electric and magnetic fields and elegantly deriving all key properties of electromagnetic radiation [31]. One might think that the structural features of electromagnetic waves, endowed with polarization and other characteristics, determine much in the operator-alphabetic foundations of cyclic and spiral bioinformatics. It is no coincidence that the alphabetic foundations of operator bioinformatics presented above turn out to be connected with the algebraic formalisms of screw calculus and quantum logic.

The vast array of inherited biological phenomena suggests the advisability of uniting both cyclic and spiral bioinformation systems into a single operator cyclo-spiral (or cyclo-screw) bioinformation system, based on a combined operator alphabet of 8 elements: $H_C$, $H_A$, $H_T$, $H_G$, $D_C$, $D_A$, $D_T$, $D_G$. For a deeper understanding of the algebraic properties of living organisms, it

seems useful to develop a cyclo-spiral (or cyclo-screw) calculus in the future as a complement to screw calculus. Additional materials on this topic are presented in the preprint [11].

## 5. Some Concluding Remarks

The importance of considering ensembles of spiral bio-antennas in organisms has already been noted above. It is appropriate to recall the topic of biological crystals (quasicrystals), whose structures allow the creation of microwave emitters, waveguides, and high-density data storage systems. Schrödinger called chromosomes aperiodic crystals [32]. The DNA double helix can be considered as a molecular crystal lattice. Important participants in the organization of molecular-genetic informatics are photonic crystals, also manifested in inherited bodily structures, for example, in the colored patterns of butterfly wings [33]. Some hybrid crystals, under the influence of a laser, behave in a manner similar to brain neurons: their reaction to light is analogous to the reaction of neurons to a stimulus [34].

In the quantum-logical approach to bioinformatics, considering the cyclic (or pulsation) features of living bodies, the author relies on a model representation of biomechanical media consisting of interconnected pulsating structures that change coherently over time. The theory of such model programmatic media can be used in the development of artificial intelligence, including in connection with the pulsating information grids (pulsars) known in computer architectures [35]. The name "pulsating" reflects the essence of this architecture, traditionally compared to the beating of a heart or pulse. Here, pulsation appears as a wave of data, and the computing process looks like the propagation of activity waves. Data entering the grid inputs begin to "pulsate" through it, transforming at each step. The grid can be configured so that different data streams collide and interact in specific cells at precisely defined cycles, generating a new "pulse" of results. This architecture is fundamentally different from the von Neumann architecture of conventional processors, as it has no central control unit; all cells operate simultaneously and synchronously; data is not stored in the classical sense but continuously "pulsates" through the processor structure, like blood flow through capillaries. The operation of such a pulsating information grid is compared to the work of the heart: the grid is likened to the myocardial muscle tissue; the processor elements to individual heart muscle cells (cardiomyocytes); the heart's beat to the electrical impulse from the sinoatrial node; computations are compared to the coordinated contraction of the heart pumping blood; information data to the pumped blood. Here, the "computation" (pumping blood) is an emergent property of the entire organ pulsating in a coordinated rhythm; no single cell is responsible for it, but all cells follow a common rhythm and local interactions. However, huge problems in programming and hardware have so far prevented pulsating information grids from becoming widespread. The most famous example of the pulsar concept is the Connection Machine, developed by Thinking Machines Corporation in the 1980s. It had up to 65,536 simple one-bit processors connected in a hypercube network. The pulsar concept is also closely related to a number of modern architectural concepts, for example, the concept of systolic arrays, whose name derives by analogy of their pulse-like operation with cardiac systoles [36].

Systolic arrays are extremely effective in artificial intelligence tasks, image processing, pattern recognition, computer vision, and other tasks with which the animal brain copes particularly well. An example is Google's Tensor Processing Unit (TPU), which uses a large two-dimensional systolic array at its core to perform massive matrix multiplications required for neural networks

with high efficiency. Cardiac systoles are not just cyclic structures, but cyclo-spiral biological structures, including the spiral movement of blood in a pulsating mode. In light of this, the task of possibly developing systolic processors by transitioning from cyclic to cyclo-spiral algorithms of operation arises.

The algebraic bioinformatics presented in this article explains the rapid evolution of organisms: complex tissues are formed not so much through the emergence of new genes, but rather through algebraic bioinformatics, whose operators change the ways existing genes are used, being linked to electromagnetic and resonance mechanisms. In light of this, the author proposes **the paradigm of "code algebraic-operator Darwinism**", according to which natural selection and the inheritance of the most survival-beneficial code combinations based on alphabetic operators of algebraic bioinformatics play an important role in evolution. One confirming example of the proposed paradigm of code algebraic-operator Darwinism is the "iron" snail *Chrysomallon squamiferum*, whose shell is made of iron and sticks to a magnet (Fig. 10). In its genome, only 11% of genes are unique. The remaining 89% are repurposed ancient genes that existed in common ancestors of mollusks more than 540 million years ago [37].

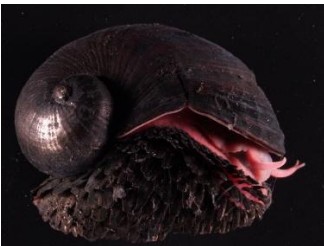

**Fig. 10** - Photograph of the "iron" snail *Chrysomallon squamiferum* from Wikipedia (https://commons.wikimedia.org/wiki/File:Chrysomallon_squamiferum_2.png); this photo is permitted for use under the Creative Commons Attribution 2.5 Generic license.

Briefly, it is noted that the formalisms of algebraic bioinformatics also allow comparative analysis of relatively short genetic sequences (e.g., DNA sequences encoding proteins), as well as consideration of genetic matrix operators that depend on time. These issues are beyond the scope of this article and deserve a separate publication. The described algebraic-operator bioinformatics seems useful for analyzing gene networks, developing digital twins, and related tasks. It is coupled with the actively developing theme of code biology [38]. Annual conferences are held on this topic, and the International Society for Code Biology operates (https://www.codebiology.org/). An important component of the formalisms of this algebraic-operator bioinformatics is tensor-unitary transformations, which provide a tensor increase in the dimension of configuration vector Hilbert spaces in quantum-logical models of growing biosystems [39].

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
