# OpenReview forum: "ALGEBRAIC BIOINFORMATICS BASED ON MATRIX-OPERATOR ALPHABETS FOR ARTIFICIAL INTELLIGENCE"
_mathai.club/MathAI/2026/Conference — 2026 Oral_

### Official Review · Reviewer_3RSC · 2026-03-10
**Accept for Track A/D (oral/poster). Strong math-AI-bio novelty; minor revision: need crarify terminology.**

**Rating:** 9
**Confidence:** 5

**Review:**

Theoretical paper proposing two matrix-operator alphabets (4 unitary Hadamard gates from DNA binary traits; related non-unitary Fibonacci/dual-number matrices) for "cyclo-spiral bioinformatics" modeling inherited cyclic/spiral biosystems, linked to quantum logic, screw calculus, AI.
1. Mathematical Rigor: high.
Formal matrices (Hadamard HnHnT=nIHnHnT=nI, unitary UU†=IUU†=I, projectors P2=PP2=P, Fibonacci powers FnFn), cyclic groups (HCn=HCn+8HCn=HCn+8), dual numbers (ϵ2=0ϵ2=0), screw calculus. Claims justified via nucleotide traits (Table 1–3), but empirical biology links are speculative; no proofs of biological efficacy.
2. Novelty & Contribution: good.
Original discovery (novel definitions): DNA binary oppositions yield Hadamard/dual/Fibonacci alphabets for "code algebraic-operator Darwinism." Bridges nucleotide bioinformatics, quantum gates, phyllotaxis, genomorphic AI. Extends author's matrix genetics; potential for tensor-unitary bio-models.
3. Relevance to MathAI: very high.
Excellent fit for Tracks A (math foundations: unitary ops, quantum logic), B (ML: genetic gates for AI), D (life sciences). Matrix algebras for bio-AI inheritance directly at math-AI intersection.
4. Technical Quality: good.
Algebra correct (e.g., DA⋅DT=F2DA⋅DT=F2, cyclic periods). Sound connections to quantum computing (Manin), screw calculus. No errors; speculative bio-applications (Mixotricha paradoxa) plausible but untested. The figures look not relevant. The copyright references are provided.
5. Clarity & Presentation: good.
Structured (intro→Hadamard→spiral→unified). Tables/figs (1–10) effective. Dense prose, typos ("ooof"?), long sentences; Russian-English translation feel; strong references (Steitz, Nielsen-Chuang).
6. AI-Generation Risk: very low.
Clearly human: idiosyncratic paradigm ("cyclo-spiral bioinformatics"), consistent author style (matrix genetics), speculative biology-physics synthesis beyond LLM patterns.
Overall Recommendation
Accept for Track A/D (oral/poster). Strong math-AI-bio novelty; minor revision: validate bio-predictions empirically, clarify AI apps (e.g., genomorphic networks).

---

### Official Review · Reviewer_zeoH · 2026-03-12
**Weakly Reject. Loose topical alignment. Better for bioinformatics venues than pure math-AI.**

**Rating:** 5
**Confidence:** 3

**Review:**

The paper proposes algebraic bioinformatics using two matrix-operator alphabets derived from DNA nucleotide properties: four unitary Hadamard matrices for cyclic bioinformatics and four non-unitary matrices (linked to Fibonacci and dual numbers) for spiral bioinformatics, aimed at modeling biological inheritance for AI applications like genomorphic AI and systolic processors.
Format is not proper (Russian text in the abstract), self-citations.
Novelty
The core idea—mapping DNA nucleotide binary traits to Hadamard matrices and related operators—is not highly novel, as the author cites their own prior preprints (e.g., on quantum-logical bioinformatics) and books, indicating this builds on established "matrix genetics" work by Petoukhov. Connections to quantum gates, screw calculus, and systolic arrays draw from known fields (quantum computing, robotics, AI hardware) without introducing new theorems or empirical validations. Score: 4/10 – Incremental originality in bio-AI framing, but largely self-referential.
Explainability
Concepts are explained through step-by-step derivations from nucleotide traits to matrices, with clear motivations from biological examples (e.g., butterfly flight, phyllotaxis spirals). However, dense prose, heavy self-citation, and speculative leaps (e.g., hidden "operator alphabets" explaining uncoordinated behaviors) reduce accessibility for mathematicians. Quantum logic and screw calculus are introduced adequately but assume familiarity. Score: 6/10 – Logical flow, but overly verbose and interpretive.
Correctness
Mathematical claims hold: Hadamard matrices are correctly defined and normalized as unitary; Fibonacci matrix powers generate Fibonacci numbers; dual number representations are standard. Biological mappings (e.g., binary traits to tables) are phenomenological and accurate to cited sources, though causal claims (e.g., matrices encoding 3D forms) lack proof or data. No errors in linear algebra or operator properties noted. Score: 8/10 – Technically sound, but biologically speculative.
References
39 references mix classics (Nielsen-Chuang on quantum computing, Birkhoff-von Neumann on quantum logic) with recent works (2024-2025 preprints, many by author or collaborators). Comprehensive for bioinformatics/quantum biology, but over-relies on self-citations (e.g., refs 11,17-19,22,39) and omits mainstream AI/math foundations critiques. All DOIs/links appear valid. Score: 7/10 – Broad but biased toward author's oeuvre.
Figures and Tables
Tables (e.g., Table 1 on nucleotide traits, Table 2 Fibonacci sequence) are simple, effective summaries of binary mappings. Figures (e.g., Fig. 1 Hadamard matrices, Fig. 4 Fibonacci powers, Fig. 9 Mixotricha paradoxa) clearly illustrate matrices, cycles, and biology, with captions explaining derivations. No clutter; visuals aid algebraic-biological links. Score: 8/10 – High quality, relevant, well-labeled.
Relevance to MathAI
MathAI focuses on mathematical foundations of AI (e.g., algebra, logic, optimization in ML/quantum AI). The paper uses linear algebra (unitary groups, projectors), quantum logic, and operator semigroups for bio-inspired AI (systolic arrays, pulsars, tensor transformations), with keywords like "quantum informatics" and "genomorphic AI." However, primary emphasis is biological modeling, not core AI theory/algorithms. Score: 5/10 – Tangential; math is sound but biology-centric.
Suitability for Publication
The work offers intriguing matrix-based bio-AI bridges but lacks rigorous proofs, experiments, or AI benchmarks, resembling a position paper more than conference-ready research. Self-promotional tone and speculation (e.g., "code algebraic-operator Darwinism") detract from rigor. Better for bioinformatics venues than pure math-AI.

---

### Decision · Program_Chairs · 2026-03-14

**Decision:**

Accept (Oral)

**Comment:**

Dear Author(s),

On behalf of the Program Committee of the International Conference on Mathematics of Artificial Intelligence (MathAI 2026), we are pleased to inform you that your paper has been accepted for an oral presentation at MathAI 2026.

Your paper was evaluated through a rigorous two-stage review process involving both automated screening and expert review by members of the Program Committee. The reviewers recognized the quality and contribution of your work.

Presentation details:

- Format: Oral presentation (15–20 minutes + 5 minutes Q&A)
- Mode: You may present either in person (offline) at the conference venue in Sirius, Russia, or remotely via Zoom. Please indicate your preferred mode when confirming your participation.
- Conference dates: Marh 30 - April 3, 2026
- Website: https://mathai.club

Next steps:

1. Please confirm your participation and presentation mode by replying to this email mathai.club@yandex.ru no later than March 15, 2026 18:00 Moscow time.
2. If you plan to attend in person, the organizing committee will provide accommodation details separately.
3. Please prepare your final camera-ready manuscript according to the formatting guidelines available at https://mathai.club and upload it to OpenReview by March 15, 2026 18:00 Moscow time.

Should you have any questions regarding the program, logistics, or your presentation slot, please do not hesitate to contact us.

We look forward to your contribution to MathAI 2026.

With kind regards,

MathAI 2026 Program Committee
International Conference on Mathematics of Artificial Intelligence
https://mathai.club
OpenReview: https://openreview.net/group?id=mathai.club/MathAI/2026/Conference
Telegram: https://t.me/MathAI_club
Email: mathai.club@yandex.ru